# Association of the past epidemic of *Mycobacterium tuberculosis* with mortality and incidence of COVID-19

Kazuo Inoue[1], Saori Kashima[2]*

1 Department of Community Medicine, Chiba Medical Center, Teikyo University School of Medicine, Ichihara, Chiba, Japan, 2 Environmental Health Science Laboratory, Graduate School of Advanced Science and Engineering, Hiroshima University, Higashi-Hiroshima, Hiroshima, Japan

* kashima@hiroshima-u.ac.jp, saori.ksm@gmail.com

**Data Availability Statement:** All relevant data are within the paper and its Supporting information files.

**Funding:** The authors received no specific funding for this work.

## Abstract

The coronavirus disease 2019 (COVID-19) pandemic, caused by the severe acute respiratory syndrome coronavirus 2 (SARS-CoV-2), has created a remarkable and varying impact in every country, inciting calls for broad attention. Recently, the Bacillus Calmette-Guérin (BCG) vaccination has been regarded as a potential candidate to explain this difference. Herein, we hypothesised that the past epidemic of *Mycobacterium tuberculosis* (*M. tuberculosis*) may act as a latent explanatory factor for the worldwide differences seen in COVID-19 impact on mortality and incidence. We compared two indicators of past epidemic of *M. tuberculosis*, specifically, incidence (90 countries in 1990) and mortality (28 countries in 1950), with the mortality and incidence of COVID-19. We determined that an inverse relationship existed between the past epidemic indicators of *M. tuberculosis* and current COVID-19 impact. The rate ratio of the cumulative COVID-19 mortality per 1 million was 2.70 (95% confidence interval [CI]: 1.09–6.68) per 1 unit decrease in the incidence rate of tuberculosis (per 100,000 people). The rate ratio of the cumulative COVID-19 incidence per 1 million was 2.07 (95% CI: 1.30–3.30). This association existed even after adjusting for potential confounders (rate of people aged 65 over, diabetes prevalence, the mortality rate from cardiovascular disease, and gross domestic product per capita), leading to an adjusted rate ratio of COVID-19 mortality of 2.44, (95% CI: 1.32–4.52) and a COVID-19 incidence of 1.31 (95% CI: 0.97–1.78). After latent infection, *Mycobacterium* survives in the human body and may continue to stimulate trained immunity. This study suggests a possible mechanism underlying the region-based variation in the COVID-19 impact.

## Introduction

The ongoing novel coronavirus disease 2019 (COVID-19) pandemic, precipitated by the severe acute respiratory syndrome coronavirus 2 (SARS-CoV-2), while a major concern, remains a mystery that puzzles the world. One part of the enigma is the remarkable and varying impact on mortality and incidence in each country. For example, the total number of

**Competing interests:** The authors have declared that no competing interests exist.

confirmed deaths due to COVID-19 per million people by country ranged from less than 0.1 to almost 1,000, indicating a four-digit difference (as of 5 April 2020) [1]. Non-biomedical factors such as socioeconomic status, cultural, and public health ones may not fully explain this vast divergence.

Currently, the Bacillus Calmette-Guérin (BCG) vaccine has gained increasing attention for its speculated efficacy against COVID-19 [2–5], which is based on the concept of trained immunity [6]. Studies have focused on the different effects of COVID-19 between countries wherein BCG vaccination programmes are currently in place and countries wherein they are not. However, one crucial perspective remains to be discussed before the argument over BCG vaccination.

Since ancient times, tuberculosis continues to pose a major threat to health worldwide [7]. It has been shown that BCG vaccination in neonates and infants reduce the risk of tuberculosis by over 50% across various populations [8]. Owing to the heavy disease burden imposed by tuberculosis, BCG vaccination is currently enforced in many countries. This is also the case for Japan, which has been battling against tuberculosis for a long time until now [9]. People are infected with tuberculosis during their youth. In 1950, more than half of the adolescents aged <20 years were infected with tuberculosis in Japan [10]. The majority (85–90%) of such infected individuals do not develop clinical manifestations of tuberculosis; however, they have latent tuberculosis infection. Even in 2018, about one-quarter of the world's population had latent tuberculosis infection [7].

In latent infection, the tuberculosis bacterium persists and continues to activate the immune system. Notably, the immunity acquired naturally due to active infection always outstrips the immunity acquired artificially from immunisation. If the 'trained immunity' by BCG vaccination exists [6], then the immunity acquired by natural infection with *Mycobacterium tuberculosis* (*M. tuberculosis*) would be more lasting than the immunity acquired artificially from BCG vaccination. Noteworthy, in the early twentieth century, tuberculosis was on the rampage of prevalence in East Asian countries including Taiwan, Thailand, Vietnam, and Japan, wherein the cumulative mortality rates for COVID-19 were remarkably lower than those in other countries such as Spain, Italy, United Kingdom, and the United States of America (USA). In the latter countries, the tuberculosis burden, regardless of whether infection rate or mortality rate, was much lower than that in the former countries.

Thus, we have noticed the following hypothesis (Fig 1). "In countries wherein BCG vaccination is currently in place, the impact (mortality and incidence) of COVID-19 may be limited because the high-risk elderly population was infected with tuberculosis in younger days to get trained immunity. Accordingly, current BCG vaccination would indicate a spurious relationship to reflect the past tuberculosis epidemic and an epiphenomenon to produce a spurious relationship with the fewer impact of COVID-19, which is seen in some countries, including Japan." We then tested this hypothesis by comparing the cumulative mortality and incidence of COVID-19 with that of past epidemic of *M. tuberculosis* (the tuberculosis incidence of 90 countries in 1990 and tuberculosis mortality of 28 countries in 1950).

## Materials and methods

### Indicators for the impact of COVID-19

The health care level in each country might be associated with the mortality rate of COVID-19. The COVID-19 outbreak rapidly expanded from April 2020 through the African continent, wherein many countries have lack of health care equipment and staff [11]. Thus, to reduce these biases, we first examined the cumulative mortality rate of COVID-19 (total number of deaths per 1 million people) on 5 April 2020; that is, before the spread of the COVID-19

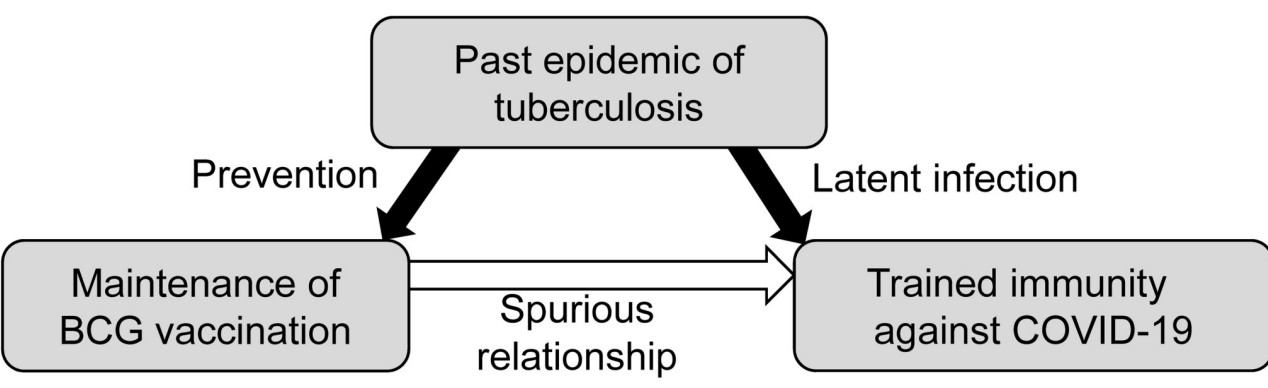

**Fig 1. The hypothesis of trained immunity in latent tuberculosis infection.**

outbreak in the African continent. Second, we examined the cumulative incidence rate of COVID-19 (total number of confirmed cases per 1 million people). We obtained data for the 98 countries that had confirmed cases of more than 200 and the death number was more than 1, on 5 April 2020, from the Our World in Data website [1]. As a supplementary analysis, we collected data on the cumulative mortality rate of COVID-19 and the cumulative incidence rate of COVID-19 on 5 August 2020, in order to evaluate the advanced situation.

### Information of the indicators of past tuberculosis epidemic for latent tuberculosis infection

We selected two indicators for the past epidemic of *M. tuberculosis*. First, we calculated the incidence rate of tuberculosis (per 100,000 people) in 1990 for 90 among the 98 countries (91.8%) from the website of Global Note as the oldest record of the incidence rate of tuberculosis [12]. Because the neonates in 1990 became 30 years old in 2020, we assumed that the rate might indicate the past epidemic context of tuberculosis that the young and middle-aged experienced. Second, we determined the mortality rate of tuberculosis (per 100,000 people) in 1950 for 28 among the 98 countries (29.2%) from four references [13–16]. Because a person born in that year is now (in 2020) 70 years old, we assumed that the mortality rate might indicate the past epidemic context of *M. tuberculosis* that the older aged experienced. The mortality data of China in 1950 were substituted by those of Taiwan. Data of South Korea during 1950–1953 had been estimated based on data in 1954 due to the Korean War [17], with an estimated mortality rate of '300 to 400 per 100,000 populations'. Therefore, the average value of 350 was used for the analysis.

### Categorisation by BCG vaccination status

We classified the targeted countries into three groups according to the status of BCG vaccination in 2011, using the BCG World Atlas [18]; group A: The country, currently, has a universal BCG vaccination programme, group B: The country used to recommend BCG vaccination for everyone, but currently, it does not, and group C: The country never had universal BCG vaccination programmes. Because Norway stopped universal BCG vaccination in 2009 [19], the group was moved from groups A to B in this analysis.

### Statistics

First, for describing the association between BCG vaccination groups and the past history of incidence and mortality of tuberculosis, we calculated the boxplot of the incidence rate of

tuberculosis (per 100,000 people) in 1990 and the mortality rate of tuberculosis (per 100,000 people) in 1950 classified by the each BCG vaccination group. Second, we created scatter plots to compare two indicators of the past epidemic status of tuberculosis against COVID-19 mortality and COVID-19 incidence on 5 April 2020. Finally, to evaluate the association between the past tuberculosis epidemic status and the impact of COVID-19, we calculated the rate ratios of the COVID-19 mortality on 5 April 2020 and their 95% confidence interval (CI), per one unit decrease in the incidence rate of tuberculosis in 1990 and mortality rate of tuberculosis in 1950 (per 100,000 people), respectively, using the generalized log-linear regression model. We also calculated the rate ratios of the COVID-19 incidence for one unit decrease in the incidence and mortality of tuberculosis, respectively. We transformed both the tuberculosis incidence and mortality rates to inverse and input the value into the model. In this analysis, we first calculated the crude rate ratios of COVID-19 mortality and COVID-19 incidence. The deaths due to COVID-19 particularly occurred in people who are older and have chronic underlying diseases, and the elderly rate and prevalence of those diseases were diverse across countries [20]. Then, we adjusted the rate of people aged 65 over the most recent year available as adjusted model 1 and additionally adjusted the prevalence rate of diabetes in 2017 and the rate of death due to cardiovascular disease in 2017 (annual number of deaths per 100,000 people) as the adjusted model 2. Finally, we additionally adjusted the gross domestic product (GDP) per capita as the adjusted model 3. These potential confounders were also obtained from the Our World in Data website [1]. As a supplementary analysis, for evaluating the current situation in COVID-19 outbreaks, we evaluated the association between both indicators of past tuberculosis epidemic statuses and the impact of COVID-19 on 5 August 2020 in the same manner as that of the main analysis.

Statistical analyses were carried out using R version 4.0.2 (R Core Team, The R Foundation for Statistical Computing, Vienna, Austria).

## Results

Fig 2 shows the incidence rate of tuberculosis in 90 countries in the year 1990 (a) and the mortality rate of tuberculosis in 28 countries in the year 1950 (b) according to the status of the BCG vaccination group in 2015. Each country was classified into three groups according to the BCG vaccination status in 2011 [18]: Group A: the country currently has a universal BCG vaccination programme, group B: the country did recommend BCG vaccination for everyone, but currently, it does not, and group C: the country never had universal BCG vaccination programmes. As shown, both the incidence and mortality rates of tuberculosis increased in group A than in groups B and C. These reflect that the country in group A is the past epidemic area of tuberculosis. The Spearman's correlation coefficients between the incidence rate of tuberculosis in 1990 and the mortality rate of tuberculosis in 1950 were 0.73 (p<0.001), which indicate a strong correlation [21].

Fig 3 shows the scatterplot comparing the log-transformed incidence rate of tuberculosis in 1990 against the log-transformed cumulative mortality rate of COVID-19 (Fig 3a), and the cumulative incidence rate of COVID-19 (Fig 3b), on 5 April 2020, in 90 countries (group A: 68, group B: 17, and group C: 5). The annual incidence rate of tuberculosis in 1990 (X-axis) ranged from 5.2 per 100,000 persons in Cyprus to 453.0 per 100,000 persons in Indonesia, indicating two digits of variation. The cumulative mortality rate of COVID-19 (Y-axis) ranged from 0.05 per one million persons in India to 943 per one million persons in San Marino, indicating over four digits of variation (Fig 3a). The cumulative incidence rate of COVID-19 ranged from 2.44 per one million persons in India to 7,632 per one million persons in San Marino, indicating over three digits of variation (Fig 3b).

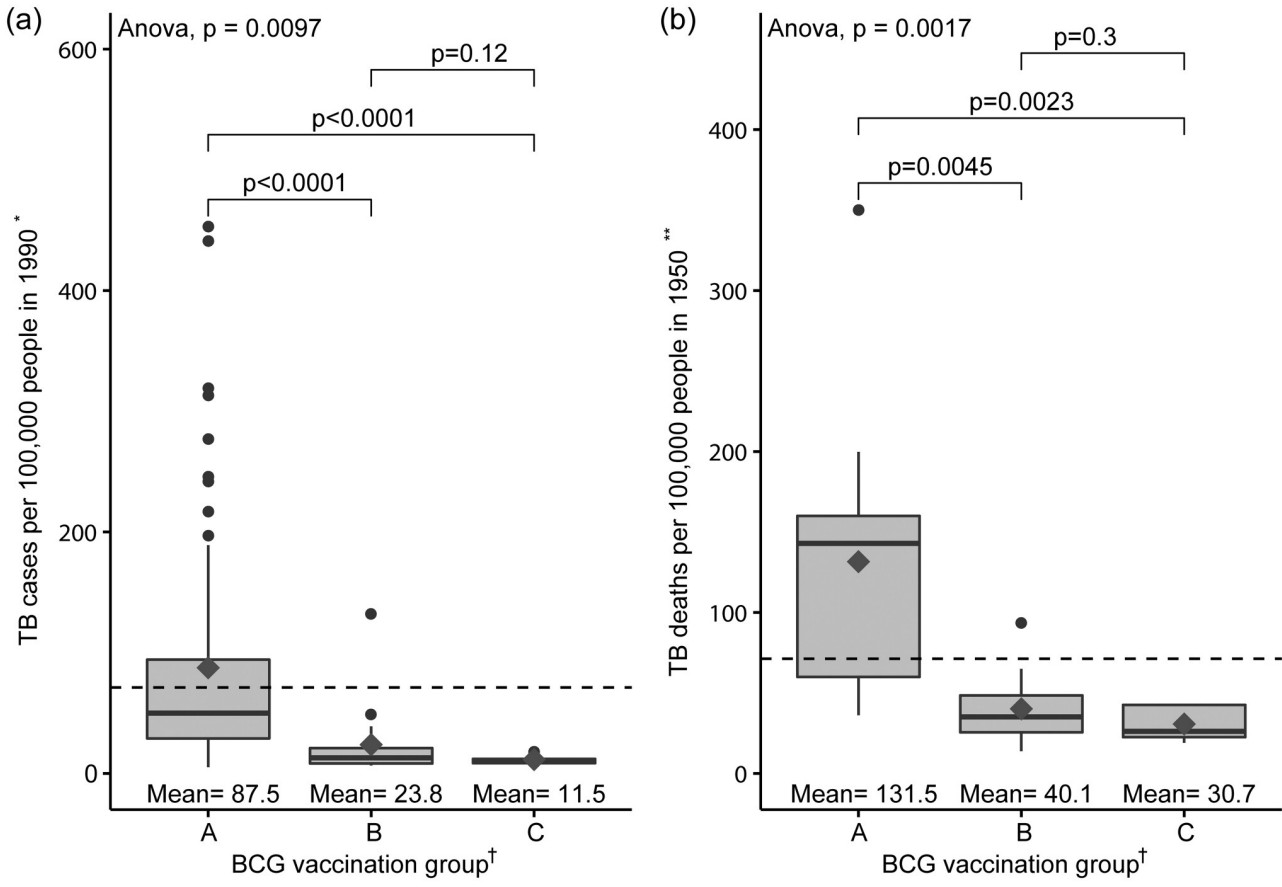

**Fig 2. (a) Tuberculosis (TB) incidence rate of 90 countries in 1990 and (b) TB mortality rate of 28 countries in 1950, according to the BCG vaccination status in 2015.** The dark grey diamond represents the mean value of the corresponding Y-axis values. The bottom and top of the box represent the 25th and 75th percentiles, respectively, and the band near the middle of the box is the 50th percentile (median). 'Whiskers' represent the maximum and minimum that extend 1.5 times the interquartile range from the box edges. The vertical dash line represents the mean of Y values across all participants in 1990 and 1950. * Information was obtained from Global Note [12]. ** Information was obtained from References [13–16]. The mortality rate of tuberculosis in China was substituted by that of Taiwan. Data of South Korea were estimated based on data collected in 1954. † Information was obtained from the BCG World Atlas [18]. Norway was moved from groups A to B [19]. The P-values shown above the bar were calculated using Bonferroni adjusted t-test. TB, tuberculosis.

The scatterplot shows an inverse relationship between the incidence rate of tuberculosis in 1990 and both COVID-19 mortality and incidence (mortality: Spearman correlation coefficient $\rho = -0.49$, $p<0.0001$; incidence: $\rho = -0.63$, $p<0.0001$). Virtually, all Asian countries that belonged to group A were located in the lower right to indicate a high incidence rate of tuberculosis and a few mortality rates of COVID-19. In contrast, most European and American countries were located in the upper left to indicate the vice versa. Here, group B (used BCG in the past) and group C (never used BCG) mixed up, with no clear difference in configuration (Fig 3a). Similar trends were also observed in the incidence of COVID-19 (Fig 3b).

Fig 4 shows the scatterplot of the mortality rates of tuberculosis in 1950 against the log-transformed cumulative mortality rate of COVID-19 (Fig 4a) and the cumulative incidence rate of COVID-19 (Fig 4b) on 5 April 2020, in 28 countries (group A: 12, group B: 11, and group C: 5). The annual mortality rate of tuberculosis in 1950 (X-axis) ranged from 13.8 per 100,000 persons in Denmark to 350.0 per 100,000 persons in South Korea to indicate one-digit variations. The cumulative mortality rate of COVID-19 (Y-axis) ranged from 0.2 per one million persons in New Zealand to 254 per one million persons in Italy to indicate three-digit

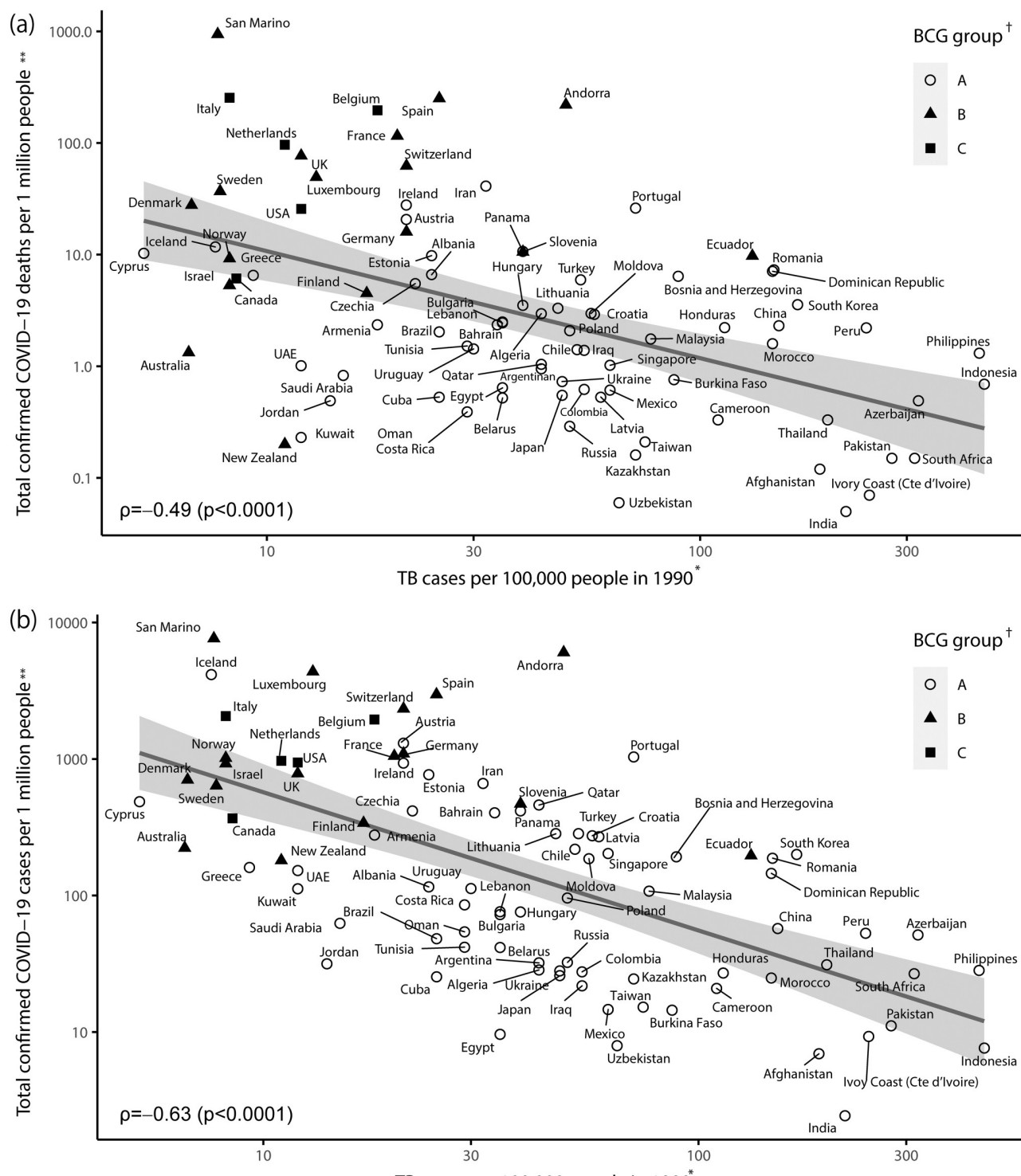

**Fig 3. Scatterplot of the incidence rate of tuberculosis (TB) in 1990 versus (a) the cumulative mortality rate of COVID-19 and (b) the cumulative incidence rate of COVID-19 on 5 April 2020 according to BCG vaccination status among 90 countries.** *Information was obtained from "Global Note" [12]. **On 5 April 2020. Information was obtained from "Our World in Data" [1]. † Information was obtained from the "BCG World Atlas" [18]. A: The country currently has a universal BCG vaccination programme. B: The country used to recommend BCG vaccination for everyone, but currently, it does not. C: The country never had universal BCG vaccination programmes. Norway was moved from groups A to B [19]. TB, tuberculosis.

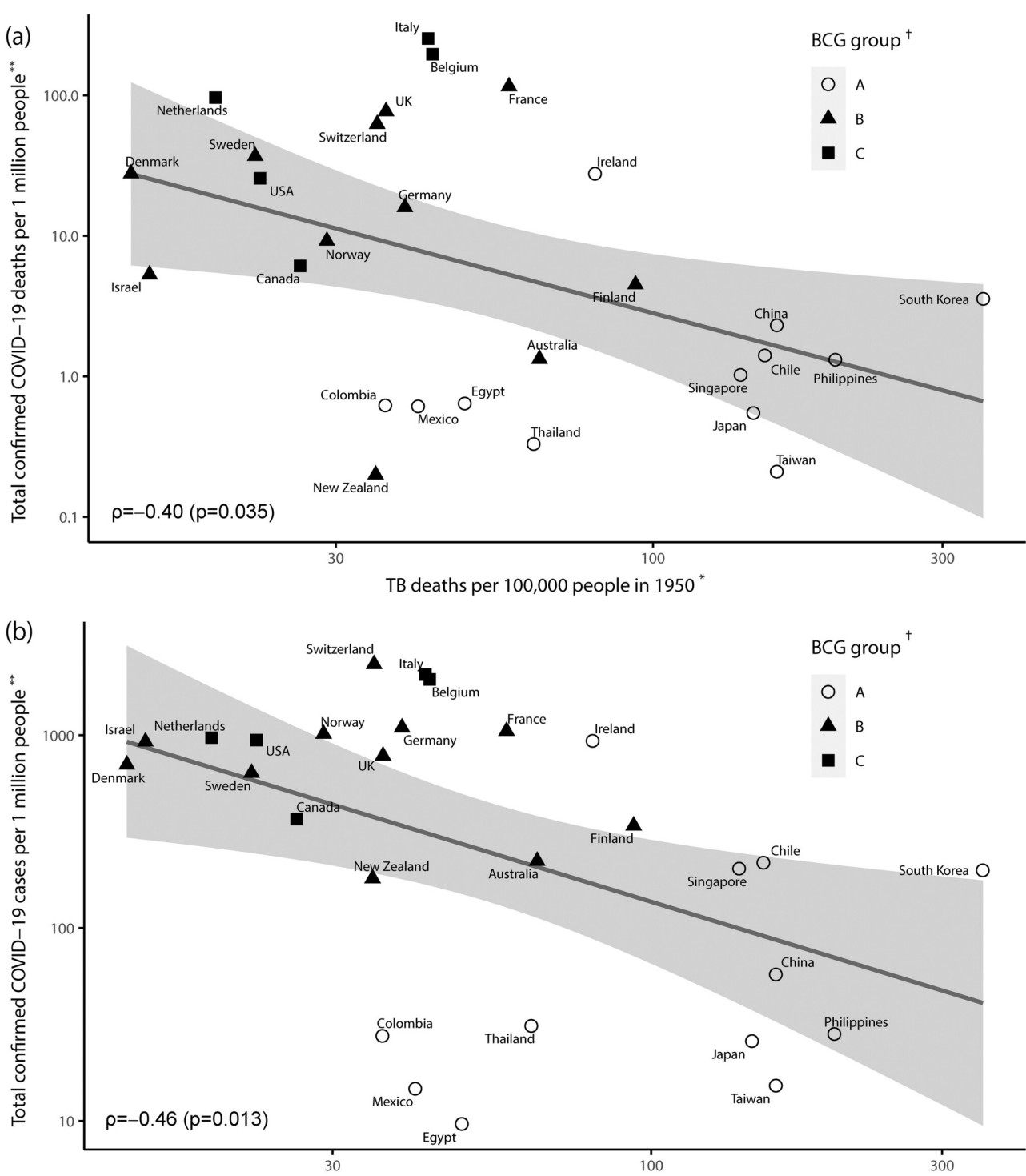

**Fig 4. Scatterplot of the mortality rate of tuberculosis (TB) in 1950 versus (a) cumulative mortality rate of COVID-19 and (b) the cumulative incidence rate of COVID-19 on 5 April 2020, according to BCG vaccination status among 28 countries.** * Information was obtained from References [13–16]. The mortality rate of tuberculosis in China was substituted by that of Taiwan. Data of South Korea were estimated based on data collected in 1954 [17]. ** On 5 April 2020. Information was obtained from "Our World in Data" [1]. † Information was obtained from "the BCG World Atlas" [18]. A: The country currently has a universal BCG vaccination programme. B: The country used to recommend BCG vaccination for everyone, but currently, it does not. C: The country never had universal BCG vaccination programmes. Norway was moved from groups A to B [19]. TB, tuberculosis.

variations (Fig 4a). The cumulative incidence rate of COVID-19 ranged from 9.62 per one million persons in Egypt to 2334 per one million persons in Switzerland, indicating a three digit variation (Fig 4b).

Similar to the relationship between the incidence rate of tuberculosis in 1990 and the impact of COVID-19, an inverse relationship was observed between the mortality rate of tuberculosis in 1950 and the impact of COVID-19 (mortality: $\rho$ = -0.40, p = 0.035; incidence: $\rho$ = -0.46, p = 0.013). Again, most Asian countries that belonged to group A were located in the lower right to indicate a high mortality rate of tuberculosis and a few mortality rates of COVID-19. In contrast, all the European and American countries were located in the upper left to indicate the vice versa. Groups B (used BCG in the past) and C (never used BCG) mixed up, as shown in Fig 3. Similar trends were also observed in the incidence of COVID-19 (Fig 4b).

Table 1 shows the rate ratios of the cumulative mortality rate of COVID-19 and the incidence rate of COVID-19 on 5 April 2020, per one unit decrease in the incidence rate of tuberculosis in 1990 and the mortality rate of tuberculosis in 1950. The rate ratio of the cumulative mortality rate of COVID-19 per 1 million was 2.70 (95% CI: 1.09–6.68) per 1 unit decrease in the incidence rate of tuberculosis (per 100,000 people) in 1990. The rate ratio of the cumulative incidence rate of COVID-19 was 2.07 (95% CI: 1.30–3.30). This association was observed even after adjusting for the rate of elders aged 65 older, country-level chronic disease prevalence, and GDP per capita [rate ratio in the adjusted model 3 for COVID-19 mortality: 2.44, (95% CI: 1.32–4.52); and for COVID-19 incidence: 1.31 (95% CI: 0.97–1.78)]. Similar trends were also observed in the association between the tuberculosis mortality rate in 1950 and the mortality and incidence of COVID-19.

In the supplementary analysis, we evaluated the association of the incidence rate of tuberculosis in 1990 and the mortality rate of tuberculosis in 1950 with the cumulative mortality rate

**Table 1. Association of the decrease in the incidence rate of tuberculosis (per 100,000 people) in 1990 and the mortality rate of tuberculosis (per 100,000 people) in 1950 with the cumulative deaths and cases of COVID-19 (per 1 million people) on 5 April 2020.**

| | TB incidence in 1990 | | TB mortality in 1950 | |
|---|---|---|---|---|
| | rate ratio [a] | (95% CI) | rate ratio [a] | (95% CI) |
| **COVID-19 mortality** | | | | |
| Crude model | 2.70 | (1.09–6.68) | 1.39 | (0.82–2.38) |
| Adjusted model 1 | 2.52 | (1.58–4.00) | 1.40 | (0.82–2.40) |
| Adjusted model 2 | 2.65 | (1.46–4.81) | 9.39 | (0.93–95.02) |
| Adjusted model 3 | 2.44 | (1.32–4.52) | 9.32 | (0.18–493.74) |
| **COVID-19 incidence** | | | | |
| Crude model | 2.07 | (1.30–3.30) | 1.59 | (0.95–2.67) |
| Adjusted model 1 | 1.73 | (1.22–2.44) | 1.61 | (1.02–2.53) |
| Adjusted model 2 | 1.42 | (1.05–1.92) | 1.54 | (0.96–2.47) |
| Adjusted model 3 | 1.31 | (0.97–1.78) | 1.49 | (0.95–2.32) |

CI, confidence interval; TB, tuberculosis

The adjusted model 1 was adjusted by the prevalence of aged 65 older.

The adjusted model 2 was adjusted by the prevalence of aged 65 older and the prevalence of chronic disease (diabetes and cardiovascular disease).

The adjusted model 3 was adjusted by the prevalence of aged 65 older, the prevalence of chronic disease (diabetes and cardiovascular disease), and the GDP per capita.

[a] Rate ratio per one unit decrease in the incidence rate of tuberculosis in 1990 and mortality rate of tuberculosis in 1950 (per 100,000 people), respectively.

and the cumulative incidence rate of COVID-19 on the later date of 5 August 2020. A scatterplot of the incidence rate of tuberculosis in 1990 against the cumulative mortality rate and the cumulative incidence rate of COVID-19 on 5 August 2020 in the 90 countries is shown in S1 Fig, while a scatterplot of the mortality rate of tuberculosis in 1950 is shown in S2 Fig. The rate ratios of the mortality rate and incidence rate of COVID-19 are shown in S1 Table. Comparing with the results of the main analysis, although the Spearman correlation coefficients were decreased in the analysis with the incidence rate of tuberculosis in 1990 (COVID-19 mortality: $\rho$ = -0.28, p = 0.008; and COVID-19 incidence: $\rho$ = -0.23, p = 0.030), a similar correlations were observed in the analysis with the mortality rate of tuberculosis in 1950 (COVID-19 mortality: $\rho$ = -0.43, p = 0.023; and COVID-19 incidence: $\rho$ = -0.40, p = 0.034). A positive increase in rate ratios of mortality of COVID-19 and incidence of COVID-19 on 5 August 2020 were also observed with decreasing the incidence of tuberculosis in 1990 and the mortality rate of tuberculosis in 1950.

The Spearman correlation coefficients between cumulative mortality of COVID-19 on 5 April and 5 August 2020 were 0.58 (p<0.0001) in the dataset of the incidence rate of tuberculosis in 1990 and 0.73 (p<0.0001) in the dataset of the mortality rate of tuberculosis in 1950, respectively.

## Discussion

In this study, high COVID-19 mortality and incidence were observed in countries that had lower experiences of a past epidemic of *M. tuberculosis*, 30 and 70 years ago. A past epidemic of *M. tuberculosis* is a strong determinant that leads to preventive policies for tuberculosis, including vaccination. Thus, trained immunity due to latent persistent infection of *M. tuberculosis* is one of the potential contributors to the low mortality and incidence of COVID-19 seen in Asian countries, contrary to that of American and European countries.

There are studies discussing the possible therapeutic effect of the BCG vaccine on COVID-19 [2–5]. However, thus far, those previous studies have not discussed the immunity acquired from natural infection by *M. tuberculosis*, which we assumed to be a leading factor against the impact of COVID-19. The results from the statistical analysis supported our hypothesis, which proposes that latent tuberculosis infection may be one of the real contributors for trained immunity, and BCG vaccination could be spuriously associated with the low impact of COVID-19 (Fig 1). In addition, one previous study reported that current high-tuberculosis-burden countries had a lower incidence of COVID-19, irrespective of the BCG vaccine status of the country, and focused on cross-immunity between *Mycobacterium* species and COVID-19 [22]. This study roughly categorized the examined countries into four groups, which are the combination of tuberculosis high and low incidence and BCG high and low coverage. Although it did not evaluate the past burden of tuberculosis, the findings may be partially consistent with ours. Asian countries that were remarked for quite a low mortality rate of COVID-19 had high levels of the past epidemic of *M. tuberculosis*. In contrast, European and North American countries, which had a high COVID-19-related mortality rate, had low levels of the past epidemic of *M. tuberculosis*. In the USA, about 60%–90% of hospitalised infected patients have comorbidities such as high blood pressure, diabetes, and heart disease or other chronic diseases, and about 80% of in-hospital mortality due to COVID-19 occurred in elderly people ($\geq$65 years) [20]. Although the prevalence rate of these factors may be diverse across countries, associations between high levels of past tuberculosis infection and low impact of COVID-19 were observed even after adjusting for indicators for country levels of elderly rate, chronic medical conditions, and national economic status. Furthermore, one study reported that pre-existing immune responses against human coronaviruses for the seasonal "common cold"

mitigate the severity of COVID-19 [23]. In addition to their finding, this study suggests that the pre-existing immune response against *M. tuberculosis* also has the potential to mitigate the disease manifestations from COVID-19. Further studies are needed to evaluate the role of pre-existing immune protection.

Low impact of COVID-19 were observed in countries receiving current BCG vaccination (group A), while a high mortality rate was observed in those not receiving it (groups B and C). Among the latter, however, there was no clear difference in the mortality of COVID-19 between group B (previously used, but discontinued BCG) and group C (never used BCG) countries. If trained immunity induced by BCG lasts until old age, then there should be a clear difference in the impact of COVID-19 in countries between groups B and C, which is in contrast to our finding. In addition, even specific acquired immunity from BCG vaccination against the incidence of tuberculosis would last only for 15 years [24, 25]. Trained immunity by natural infection with *M. tuberculosis* would last for some decades, which is far longer than that of specific acquired immunity by vaccination.

Another fact to be noted is that the infection pattern (morbidity and mortality) of tuberculosis has significantly changed in some countries, which might affect the strength of people's training immunity induced by *M. tuberculosis*. Accordingly, this may be associated with the different mortalities of COVID-19 on the population in those regions to some extent. However, most deaths from COVID-19 are older people who may have been exposed to tuberculosis at a young age. Therefore, this study primarily examined the association between past exposure to tuberculosis and COVID-19 mortality. We hypothesized that latent tuberculosis infection induces continuously trained immunity, which may explain the lower mortality due to COVID-19 found in this study. A recent animal study [26] demonstrated that *M. tuberculosis* inhibited trained immunity in contrast to BCG, which appeared to contradict our hypothesis. They experimentally administered *M. tuberculosis* to mice intravenously, which resulted in the radical systemic invasion. In fact, all *M. tuberculosis*-administered mice succumbed by 120 days. In such a special situation, it is not surprising that *M. tuberculosis* is life-threatening, including disrupted immune function. Therefore, their findings may not be directly discussed or compared to ours.

Both group B and C countries had a relatively low impact of the past tuberculosis epidemic, where maintenance of BCG vaccination has been differed by a political decision. For example, Norway terminated universal BCG vaccination in 2009, mainly due to a very low incidence of tuberculosis [19]. Therefore, although Norway had been classified as group A in 'the BCG World Atlas' in 2011 [18], the group is virtually moved from groups A to B in the analysis. It is obvious that the past tuberculosis epidemic is the strong determinant that precedes preventive policy for tuberculosis, including vaccination. Therefore, it appears that BCG vaccination is spuriously associated with COVID-19 impact, as indicated in Fig 1.

There are several limitations to this study. First, owing to the ecological study design, caution is needed to apply this finding to individual levels. Also, because of the data limitation of the past epidemic of *M. tuberculosis*, we could only obtain the data in 1950 and 1990. Ideally, further investigation is desirable to compare the progression of COVID-19 symptoms between people who had a latent infection of tuberculosis and those who do not. Second, the worldwide burden of COVID-19 has changed moment by moment, and it was rapidly expanded through the African and South American regions. In this study, eight African countries and seven South American countries were evaluated. Although we observed positive associations of the past epidemic of *M. tuberculosis* with the impact of COVID-19 and in data on 5 August 2020, the associations were being attenuated when comparing the result in data on 5 April 2020. The effects of the current health care system should also be evaluated in further studies. In addition, although we have taken into account the rate of elderly people aged 65 years, country-level

chronic disease prevalence (diabetes and coronary heart disease), and GDP per capita, other biases such as other comorbidities, differences in diagnostic and treatment of disease, differences in reporting of COVID-19, and genetic differences in the population remain unadjusted. Our hypothesis needs to be tested continuously and, in the future, examined in different stages and settings of the COVID-19 pandemic.

In conclusion, high COVID-19 mortality and incidence were observed in countries with lower experiences of the past epidemic of *M. tuberculosis*. Trained immunity has potentially contributed to the low impact of COVID-19 in Asian countries and the high impact of COVID-19 in European and American countries and may play an important role in preventing the exacerbation and death of individuals exposed to COVID-19. This study suggests that pre-existing immunity against *M. tuberculosis* is also one of these contributors. Given this, risk stratification and preventive measures from this viewpoint can be explored. Thus, our findings warrant further investigation.

## Supporting information

**S1 Fig. Scatterplot of the incidence rate of tuberculosis (TB) in 1990 versus (a) the cumulative mortality rate of COVID-19 and (b) cumulative incidence rate of COVID-19 on 5 August 2020 according to BCG vaccination status among 90 countries.**
(PDF)

**S2 Fig. Scatterplot of the mortality rate of tuberculosis (TB) in 1950 versus (a) cumulative mortality rate of COVID-19 and (b) cumulative incidence rate of COVID-19 on 5 August 2020 according to BCG vaccination status among 28 countries.**
(PDF)

**S1 Table. Association of the decrease in the incidence rate of tuberculosis (per 100,000 people) in 1990 and the mortality rate of tuberculosis (per 100,000 people) in 1950 on the cumulative deaths and cases of COVID-19 (per 1 million people) on 5 August 2020.**
(PDF)

## Author Contributions

**Conceptualization:** Kazuo Inoue.

**Data curation:** Kazuo Inoue.

**Formal analysis:** Kazuo Inoue, Saori Kashima.

**Investigation:** Kazuo Inoue, Saori Kashima.

**Methodology:** Kazuo Inoue, Saori Kashima.

**Project administration:** Kazuo Inoue, Saori Kashima.

**Resources:** Kazuo Inoue, Saori Kashima.

**Software:** Kazuo Inoue, Saori Kashima.

**Supervision:** Kazuo Inoue.

**Validation:** Kazuo Inoue, Saori Kashima.

**Visualization:** Kazuo Inoue, Saori Kashima.

**Writing – original draft:** Kazuo Inoue, Saori Kashima.

**Writing – review & editing:** Kazuo Inoue, Saori Kashima.

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
