## [Decision Letter · Decision Letter 0]

7 May 2021

PONE-D-20-38030

Association between past tuberculosis epidemic and COVID-19-related mortality through latent infection by natural immunity

PLOS ONE

Dear Dr. KASHIMA,

Thank you for submitting your manuscript to PLOS ONE. After careful consideration, we feel that it has merit but does not fully meet PLOS ONE’s publication criteria as it currently stands. Therefore, we invite you to submit a revised version of the manuscript that addresses the points raised during the review process.

I would like to apologize for the length of time taken to respond, but we had difficulty in obtaining the required number of reviewers. I hope you understand that I felt the need for your manuscript to be reviewed by two reviewers and this led to the delay. Please take a careful look at each of the reviewers' comments.

We look forward to receiving your revised manuscript.

Kind regards,

Angelo A. Izzo

Academic Editor

PLOS ONE

Journal Requirements:

Reviewers' comments:

Reviewer's Responses to Questions

**Comments to the Author**

1. Is the manuscript technically sound, and do the data support the conclusions?

Reviewer #1: Partly

Reviewer #2: Yes

2. Has the statistical analysis been performed appropriately and rigorously? 

Reviewer #1: Yes

Reviewer #2: Yes

3. Have the authors made all data underlying the findings in their manuscript fully available?

Reviewer #1: Yes

Reviewer #2: Yes

4. Is the manuscript presented in an intelligible fashion and written in standard English?

Reviewer #1: Yes

Reviewer #2: Yes

5. Review Comments to the Author

Reviewer #1: In this study, Inoue and Kashima analyze the relationship between latent tuberculosis and COVID19 mortality. An inverse association between BCG and COVID19 has been proposed by several ecological and epidemiological studies, while latent tuberculosis was much less studied. From this perspective, this is an useful survey of the data in various countries. However, such ecological studies are prone with many biases, and this needs to be very clearly stated.

Comments

1. The authors have tried to correct for age, (some) co-morbidities and GDP, but of course many biases remain in such studies, such as (but not restricted to) other co-morbidities, differences in diagnostic and treatment of disease, differences in reporting of COVID19, genetic differences in the population, etc.

2. The authors hypothesize that latent TB induces continuous trained immunity, that may explain the lower mortality due to COVID19. However, a recent study (Khan et al, Cell 2020) demonstrated that TB (in contrast to BCG) actually inhibits trained immunity, and this is counterintuitive for the hypothesis of this study. This aspect needs to be discussed.

3. While large parts of the world are exposed to TB and others not (such as developed countries), everyone in the world is continuously exposed to environmental non-tuberculous mycobacteria. Why are these non-tuberculous mycobacteria not protective as well?

4. In some countries the TB-related morbidity and mortality significantly changed during our lifetime. Is that associated with different mortalities in those segments of the population?

Reviewer #2: In this manuscript, the authors hypothesized the past tuberculosis epidemics may be one of the latent explanatory factors which can reduce the COVID-19 mortality. And they also revealed the BCG vaccination which could modulate or train natural immunity and protect COVID-19 severity as spurious association with regional tuberculosis epidemics .

Specific recommendations for revision

a)Major:

Recently the reverse relation of national or regional tuberculosis incidence and BCG coverage with COVID-19 incidence (M.Madan, S.Pahuaja, A.Mohan, et.al https://doi.org/10.1016/j.puhe.2020.05.042) was found and

other researchers mention that BCG can reduce the incidence of COVID-19 (Melvin Joy, B. Malavika, Edwin Sam Asirvatham et.al(https://doi.org.10.1013/jcegh.2020.08.015) The authors might

mention about the association of past tuberculosis epidemic and COVID-19 not only mortality but also incidence. And the authors could show the spurious relationship of BCG vaccination and trained natural immunity against COVID-19 via past epidemic of tuberculosis.

b)Minor

Page2 line17 should be written precisely as The coronavirus disease 2019(COVID-19) caused by SARS-COV2

Page2 line18 each region may be written each region on earth

Page2 line19 may paraphrase to explain the difference in mortality

Page2 line30 trained immunity may rewrite trained natural immunity

Page3 line51 new-born may be newborn

Page5 line 102 ditto

Page15 line344 immune may be immunity

6. PLOS authors have the option to publish the peer review history of their article (what does this mean?). If published, this will include your full peer review and any attached files.

Reviewer #1: No

Reviewer #2: **Yes: **Mustuo Shibata

---

## [Author Response · Author response to Decision Letter 0]

27 May 2021

We much appreciate your kind and thoughtful comments on our manuscript. Following the reviewer’s helpful comments, we have carefully revised our manuscript. We have sent our response letter as the attached file "Response to Reviewers".

---

## [Decision Letter · Decision Letter 1]

1 Jun 2021

Association of the past epidemic of Mycobacterium tuberculosis with mortality and incidence of COVID-19

PONE-D-20-38030R1

Dear Dr. KASHIMA,

We’re pleased to inform you that your manuscript has been judged scientifically suitable for publication and will be formally accepted for publication once it meets all outstanding technical requirements.

Kind regards,

Angelo A. Izzo

Academic Editor

PLOS ONE

Additional Editor Comments (optional):

Reviewers' comments:

Reviewer's Responses to Questions

**Comments to the Author**

1. If the authors have adequately addressed your comments raised in a previous round of review and you feel that this manuscript is now acceptable for publication, you may indicate that here to bypass the “Comments to the Author” section, enter your conflict of interest statement in the “Confidential to Editor” section, and submit your "Accept" recommendation.

Reviewer #1: All comments have been addressed

Reviewer #2: (No Response)

2. Is the manuscript technically sound, and do the data support the conclusions?

Reviewer #1: Yes

Reviewer #2: (No Response)

3. Has the statistical analysis been performed appropriately and rigorously? 

Reviewer #1: Yes

Reviewer #2: (No Response)

4. Have the authors made all data underlying the findings in their manuscript fully available?

Reviewer #1: Yes

Reviewer #2: (No Response)

5. Is the manuscript presented in an intelligible fashion and written in standard English?

Reviewer #1: Yes

Reviewer #2: (No Response)

6. Review Comments to the Author

Reviewer #1: The authors have responded appropriately to my comments. I have no additional suggestions.

Reviewer #2: I am fully satisfied with your responses to my comments. I think your manuscript is acceptable and this paper will stimulate further investigations on trained immunity against SARS-Co-V2 and the region-based variations of the COVID-19 impact.

7. PLOS authors have the option to publish the peer review history of their article (what does this mean?). If published, this will include your full peer review and any attached files.

Reviewer #1: No

Reviewer #2: **Yes: **Mutsuo Shibata

---

## [Editor Report · Acceptance letter]

10 Jun 2021

PONE-D-20-38030R1 

Association of the past epidemic of *Mycobacterium tuberculosis* with mortality and incidence of COVID-19 

Dear Dr. KASHIMA:

I'm pleased to inform you that your manuscript has been deemed suitable for publication in PLOS ONE. Congratulations! Your manuscript is now with our production department. 

Kind regards, 

on behalf of

Dr. Angelo A. Izzo 

Academic Editor

PLOS ONE